# A Power-Line Communication System Governed by Loop Resonance for Photovoltaic Plant Monitoring

**DOI:** 10.3390/s22239207

**Published:** 2022-11-26

**Authors:** José Ignacio Morales-Aragones, Matthew St. Michael Williams, Halleluyah Kupolati, Víctor Alonso-Gómez, Sara Gallardo-Saavedra, Alberto Redondo-Plaza, Miguel Ángel Muñoz-García, Francisco José Sánchez-Pacheco, Luis Hernández-Callejo

**Affiliations:** 1Departamento de Física, Universidad de Valladolid, 42004 Soria, Spain; 2Physics Department, University of the West Indies, Kingston 11000, Jamaica; 3Electrical, Electronics, and Computer Engineering Department (EECE Department), University of Pretoria, Lynnwood Road, Hatfield, Pretoria 0002, South Africa; 4Departamento Ingeniería Agrícola y Forestal, Universidad de Valladolid, 42004 Soria, Spain; 5Departamento Ingeniería Agroforestal, Universidad Politécnica de Madrid, Av. Puerta de Hierro 2, 28040 Madrid, Spain; 6Departamento Tecnología Electrónica, Universidad de Málaga, Avda. Cervantes 2, 29071 Málaga, Spain; 7ADIRE-ITAP, Departamento Ingeniería Agrícola y Forestal, Universidad de Valladolid, 42004 Soria, Spain

**Keywords:** low cost, monitoring, power-line communication (PLC), resonance

## Abstract

Within this paper, a PLC system that takes advantage of the loop resonance of an entire DC-PV string configured as a circular signal path is developed and implemented. Low cost and extremely simple transceivers intended to be installed within each PV module of a string have been designed and successfully tested. In addition, an anti-saturation coil has been conceived to avoid saturation of the core when the entire DC current of the string flows through it. Bi-directional half-duplex communication was successfully executed with up to a 1 MHz carrier frequency (150 kbps bitrate), using a simple ASK modulation scheme. The transmission and reception performance are presented, along with the overall system cost in comparison to the previous literature.

## 1. Introduction

An overwhelming amount of confidence is placed within the potential of renewable energy (RE) systems to meet the global demand necessary to thwart the occurrence of additional cumulative greenhouse gas emissions (GHG), induced by fossil fuel use. This is of course envisioned to occur through the rapid erection of new renewable installed capacity and the decommissioning of conventional power plants. Photovoltaic (PV) solar, in particular, has seen a significant rise in installation as a function of the steadily decreasing costs of the technology. For instance, an 85% cost reduction in global utility-scale PV levelized cost of electricity (LCOE) between 2010 and 2020 was observed, corresponding to a 1600% (42 GW:2010, 714 GW:2020) cumulative increase in installed capacity [1]. Comparably within the same decade, onshore wind displayed a cumulative increase in capacity of 293% (178 GW:2010, 699 GW:2020). This trend evokes great prospects for continued global PV proliferation and warrants the interest that has been seen among studies to uphold the potential of the technology through the development of multiple PV monitoring systems and algorithms, as documented in [2,3,4]. Fundamentally, ensuring the health of PV plants are preserved as they continue to gain significant prominence, especially with their observed disposition toward a variety of physical faults and failures [2].

Amongst the different PV monitoring methodologies, solar module level I–V (current-voltage) curve tracing was utilized in the genesis of our precursory work to this paper, not only due to the accuracy of the method but also in large part because of the ability to perform the method with integrated low-cost electronics [4].

However, despite the demonstration of the feasibility of our preliminary monitoring system, further cost reduction is always necessary to ensure system affordability is universally flexible across the widest range of consumers or socioeconomic groups. The functionality of monitoring systems can aid with building long-term consumer trust in the RE industry by bolstering the general outlook of RE technologies as reliable, long-term investments. As such, it is paramount that continuous, cost reduction be applied to these systems to incentivize the consumer base to integrate them with their PV installations. The communication block for I–V tracing is a section that can be designed to further regulate the cost reduction of the system. Several studies have utilized differing wired and wireless communication strategies assessing their cost, transmission speed, and efficiency. Ref. [3] presented a brief review of some of these studies and highlighted the advantages and limitations found among them. The review communicates that wired methods such as coaxial cables and fiber optics, present high transmission rates with low attenuation but are ascribed with limitations in coverage and expense respectively. Wireless methods such as Satellite, Wi-Fi, Bluetooth, GSM, Radio-Frequency transmission, and others are either characterized by fast transmission rates plagued by high expense and frequency regulations, low-cost with limitations due to interference effects, and transmission distance, or high expense and slow transmission.

Refs. [3,5] report the popular rationale for the use of the wired strategy of power-line communications (PLC) in PV monitoring. PLC systems in PV environments can provide transmission speeds up to 200 Mbps while subverting the costly manner of several wired and wireless methods due to their innate topology of using the same DC-PV power lines as a transmission medium. The following section will review the current PLC configurations found in literature. In [6] the authors proposed a frequency shift-keying (FSK) method at a carrier of 132.5 kHz with the use of the commercial ST7540 FSK power line transceiver (with an internal power amplifier) and Microchip PIC18LF4620 microcontroller to transmit along a 10-module series string. The PLC signal showcased negligible interference with the solar module MPPT operating points and inverter voltage whilst being properly detected. The PLC board of the system was however at the time of the study, 30 euros per board.

Ref. [7] utilized a pulse modulation methodology at low frequency (1 kHz) for PLC communication, by using a switchable current source. The authors tested the method among four different self-built transmitter configurations and used a current transformer for sensing at the receiver, housed at the inverter. Each transmitter produced a distinct pulse and showed no alteration in features when the I-V operating point was varied, however varying PV array configurations resulted in signal attenuation either by increasing the number of modules in series (20 × 1) or the number of strings in Parallel (5 × 4, 10 × 2). This performance was relative to the authors’ reference configuration of 10 solar modules in series (10 × 1). Whilst the signal strength was still favorable for the authors, the communication speed was low (about 1 kbps). Ref. [8] employed base band communication under CENELEC B band (95–125 kHz) with capacitive coupling to the power line. They used bypass capacitors across the PV modules but no AC blocking coil to prevent the communication signal from flowing through the modules. The transmission was driven by a Texas Instruments MSP430 microcontroller at 100 kHz along a simulated 1000 m string, with an inline line trap attached to filter the carrier signal to reduce harmonics at the inverter input. The PLC module configuration approximated to about 10 euros or 2% the cost of a 250 kWp solar module, significantly cheaper to [6]. The authors estimated that using RS-232/RS-485 wired or radio wireless topologies would equate to 30 and 18 euros more respectively. Ref. [5] utilized a PLC system driven by commercial PLC components as in [6], which was governed by orthogonal frequency division multiplexing (OFDM) modulation. With this setup, the authors could obtain transmission speeds up to 4000 kbps as stated in [9], due to the high-frequency band (2–30 MHz) capabilities of the components, but as in [6], the use of commercial components was expensive [10,11,12]. All employed PLC communication with Amplitude Shift-Keying, with [10] describing some basis for the design of a modem to employ direct current (DC) PLC. In [11], the authors proposed a grouping method for a residential PV system of four solar modules per PLC module to assess the possible cost reduction in comparison to their preliminary system of one solar module per PLC module in [9]. The new configuration gave rise to a 72% cost reduction; however, with this setup voltage becomes the only differentiating variable at the solar module level. Their PLC system was driven by a 16-bit MCU with the use of capacitive coupling as in [8]. In [12] the PLC system was dictated by parallel resonant coupling through LC resonators and driven by a 32-bit MCU. The system was tested along a 16 solar module string, with modules each of 255 W. A 200 kHz pulse width modulated (PWM) carrier wave was generated and used from the MCU. The wave was transformed to a sinusoidal waveform through an m-derived low pass filter designed to filter three levels of harmonics. The MCU then applied digital switching to modulate the wave. Within this study, the authors demonstrated a sizable increase in transmission efficiency when using their coupling circuit in comparison to conventional capacitive coupling as found in [8,9]. This gain in performance was especially identifiable as the impedance of the coupling circuits were increased. The authors also compared the output signal strength of both coupling methods by validating against the open circuit output signal of a coupling circuit. Within this test, very little signal reflection/attenuation was recorded during transmission when using parallel resonance relative to capacitive coupling. Above all else, the estimated cost of the PLC board equated to 5 USD/4.52 EUR per board. However, in this case the DC power current of the string is forced to flow across the resonance coil and depending on its value the coil core can be pushed to saturation and the resonance condition will not be fulfilled. The solution to this core saturation problem will be one of the novelties presented in our work. Additionally, the authors showcased the importance of proper detection firmware, as this heightened tolerance of their PLC system to bit error rates (BER). In [13] the authors propose the integration of the PLC with dc-dc power optimizers (DCPO), where the communication signal is added in the power control loop of the DCPO in such a way that the data are modulated at its output and transmitted via the cascaded power line. AC Voltage sampling circuits are installed in DCPOs for reception. Ref. [14] used FSK as in [6] but with 110 kHz representing a logic one and 90 kHz a logic 0; capacitive coupling was used with an AC blocking coil.

Lastly, in [15], PLC transmission was used to facilitate an integrated fault detection algorithm in the communication module which deciphers the specificity of the PV fault occurring at a 10 min interval along an 18 module PV array. In totality, all the studies highlighted the robust nature of PLC systems in their simplicity, efficiency, and cost. As such this paper will explore the implementation of the ring PLC configuration we proposed in [16] which takes advantage of resonance along the entire DC-PV line (loop resonance) for medium to high-frequency communication. With this topology the line essentially acts as a resonant loop antenna with high Q-factor characteristics for maximum power transfer between transmitter and receiver. This paper will evolve with the following sections: Section 2 Materials and Methods, Section 3 Results, and Section 4 Conclusions.

## 2. Materials and Methods

Within this section, all configurations of passive and active components used to develop the PLC system and propagate its waveform along the PV-DC loop are discussed. The location and PV system where the PLC system was mounted, is the same as that detailed in [4]; a grid-connected eleven module array with varying defects.

### 2.1. Loop Resonance

A PLC system dictated by loop resonance, becomes attributed with the observed tendency of one-wavelength loop resonant antennas, to maintain high levels of voltage/current signal amplitudes along each point of its full length. This allows for consistent carrier power across each point along the loop between the transmitter and receiver. The loop when tuned presents a high Q-factor at resonance which allows itself to act as a narrowband filter to filter out-of-band signal noise around its resonance frequency.

In order to implement our ring-shaped communications signal path, we propose to install a bypass capacitor across the terminals of the combiner box which closes the series PV string line in a loop for AC signals. Our previous work [16] illustrates the theoretical approach derived for loop resonance communications and presents the implementation of the solution towards the problem of dynamically retuning the loop, when facing inevitable changes that could affect the loop resonance condition.

In addition, we have designed low-cost communication electronics connected across each solar module leads with a series coil that blocks the AC signal from flowing across the module as shown in Figure 1. This will avoid the possibility of a very low impedance path through the module (depending on its operating point) that would short circuit the communication signal. This low impedance path will occur for a module working in the right section of the I-V curve, where the dynamic impedance is effectively very low. Capacitive coupling is used to inject the communication signal into the loop from the designed transceiver.

### 2.2. Coil Saturation

Most of the configurations presented before make use of coils (or transformers) in series with the DC current path of the PV installation. These components show a serious limitation related to the saturation of the core material under high DC currents that make them stop behaving as coils or transformers for the communications signals. Increasing currents in modern solar modules will make these configurations ineffective at high irradiances. To overcome this drawback for our blocking coil we have developed an anti-saturation coil composed of two equal bindings in series winded in opposite directions that lead to a zero flux inside the core for DC currents, avoiding its saturation. To allow inductive behavior for AC signals, a capacitor C is placed in shunt configuration with one of the bindings as shown in Figure 1 (L1). Network analysis of this configuration shows a device impedance: $Z=j\frac{\omega L\left(1-K\right)\left(\omega^{2}LC\left(1+K\right)-2\right)}{\omega^{2}LC-1}$ (where C is the capacitance of the capacitor attached to the coil, L is the autoinduction coefficient of each of the two bindings and K is the coupling coefficient between them). For high enough frequency AC signals where $\omega^{2}LC\left(1+K\right)\gg 2\;and\;\omega^{2}LC\gg 1$, Z can be expressed as Z=jωL(1−K)(1+K) and the device shows the inductive behavior desired with an equivalent autoinduction coefficient equal to $L_{equ}=L\left(1-K^{2}\right)$.

### 2.3. Electronics Designed

The central hub of the PLC configuration was the PIC16F1615 8-bit Microcontroller (MCU). A simple ASK modulation scheme was used. For a minimum cost and for simplicity an integrated comparator within the microcontroller has been used as a reception device. For transmission, the switching mechanism for modulation was derived from the MCU working in conjunction with the TSM8588CS_A1811 push-pull MOSFET integrated circuit (IC), with an n-MOSFET and a p-MOSFET encapsulated together. This way, a minimum number of external components are needed, and the main working load relies on the microcontroller and its firmware.

Figure 1 shows the schematics of the TX–RX electronics. The microcontroller is placed in the center, with the detail of its integrated comparator along its two analog inverting and non-inverting inputs, and the two digital outputs used for driving the TX power amplifier. Regarding the TX section on the right of the figure, the two digital outputs drive the MOSFET gates. The two MOSFETs are connected in a push-pull configuration between the positive and negative leads of the solar module, so a TX amplitude equal to the module voltage can be achieved. Two resistors (R5 and R6) between gate and source provide a way for gate charge evacuation, improving the switching speed. The n-MOSFET is driven directly from the microcontroller output since the switching thresholds of the MOSFET are digitally compatible. In turn, the p-MOSFET must be driven from the second digital output through a capacitor, because a DC voltage displacement is needed to trigger the p-MOSFET gate. The digital independent control of the two gates allows for the ease in programming within the firmware all the output signal parameters (frequency, pulse duration, dead times) while specially preventing shot-through during push-pull operation. The value of the two resistors R7 and R8 allows for adjusting the output power sent to the line and determining the output impedance of the transmitter. Finally, the capacitor C3 prevents the DC voltage from the string line to enter the TX power stage. Regarding the RX section on the left, the resistor divider (R1, R2 and R3) provides a bias DC voltage to the comparator input representing a threshold for the RX signal that will avoid spurious switching of the comparator because of the noise at the input line. RX signals are injected from the line through the DC isolating capacitor C2 and resistor R4 to the input resistor R2. Capacitor C1 bypasses resistor R2 for AC signals, improving the input sensibility. When the amplitude of a received signal is over the threshold mentioned, the output of the comparator switches according to the RX signal, and its output can be read by the firmware in a digital fashion and processed for demodulation. The anti-saturation coil (L1) designed and described before is shown also in Figure 1 placed between the positive lead of the String and the positive lead of the solar module, avoiding the flow of the communications signal through the module, and making the performance independent of its working point.

No filtering of the output signal has been implemented, because as proved in our field measurements the high Q resonance characteristics of the loop filter properly rejects harmonics out of the carrier frequency chosen.

Our general setup for field measurements is sketched in Figure 2, where a PV string is shown with several blocks in series labelled “A”. Each “A” block represents one PV module in the string with its communications electronics and anti-saturation coils connected as detailed in the same figure. In the combiner box the “A” block has no PV module connected but just a short circuit in place to allow DC current flow, and in addition in this case the serial port integrated within the microcontroller is activated to receive external commands from a host and to send back diagnostics data to it. Anti-saturation coils have been also connected as AC blocking devices in the combiner box avoiding communication signals to enter the DC path outside the string. These coils also block high frequency harmonics from inverters or other devices in the DC output line to enter the communications loop. The DC power current path is colored blue, and the communications AC signal is colored red. The AC voltage generator (Vac) within each “A” block is the circuit equivalent to the power push-pull stage for transmission. Note that the negative side of the communication electronics is connected to the negative lead of the solar module and is a sort of local ground for this module circuitry. As stated in [6] no common reference voltage exists for the whole string due to its configuration, and to avoid confusion the “ground” symbol and name are not used in our schematics, since the local ground of one module (negative) is the positive lead of the previous module. This circumstance allows us to regard the circular communications path as a serial association of voltage generators, where only one of them is driving the signal into the loop while others show an input impedance that leads to a voltage drop for communications signals, enough for reception to work. The system is intended to work in an answer-response fashion in such a way that the transmitter in the combiner box initiates communication with a request sent to one of the receivers that listens for everyone along the loop. The first byte sent is the unique identification address of the receiver requested, that after receiving its address and subsequent command, will respond with the data requested. All parameters of the circuitry and firmware have been carefully adjusted to optimize the communication even in the presence of relatively high levels of noise. When the combiner box is transmitting, the power sent to the loop must be divided equally for the rest of transceivers along the loop. Therefore, all of them must show the same (relatively low) impedance, but for the inverse communication, the reception level can be improved since only the combiner box electronics must receive the response signal, and it can be programmed to show high impedance during reception. This way, most of the signal transmitted back by the requested module will reach the combiner box and only a small fraction of it will be lost along the rest of the loop. Considering that the most volume of information is expected to flow from PV modules toward the combiner box, this last strategy will minimize considerably the bit error probability (BER).

The loop resonance frequency of the closed signal path established will depend on the length of the path and on the series resistance, inductance and capacitance found along the AC loop as stated in [16], and our loop is supposed to always fulfill the resonance condition, including, if necessary, an automatic tuning circuitry as the one designed in [16].

## 3. Results

Transmission tests have been performed over the setup described before within our 11-module PV plant, with two communication circuits as the one shown in Figure 1 connected in one of the PV modules and in the combiner box respectively. In the rest of the PV modules, a simple circuit is replicated and attached to each to simulate the total impedance in the reception mode of the communications electronics, with the corresponding anti-saturation coils connected, in order to get the behavior of the entire loop as close as possible to the real setup. The combiner box bypass capacitor, was adjusted in such a way that after connecting the transceivers and impedance simulating circuits, the loop resonance frequency of the signal path was 1 MHz. All the measurements have been performed with the inverter on and the plant working (with a DC string current around 5 Amps) in order to test the anti-saturation coil performance.

A test signal composed of 1 MHz carrier bursts was injected in the loop by one of the transceivers in TX mode. Two different Bursts were alternated in time seven and three carrier cycles long respectively, representing a digital modulation of alternating 0,s and 1,s. The dead time between bursts is programmed for a fixed symbol duration of 13 us, same for “0” and “1”. The signal received in the circuit on reception mode was measured with an oscilloscope and is shown in Figure 3a. The ability of the push-pull TX stage of featuring a high output impedance by opening both n and p MOSFETS allows for the shortening of the damped resonance oscillation after the end of each burst, avoiding prolongation of the symbol transmitted and the consequent Inter Symbol Interference (ISI), as seen in Figure 3a.

Figure 3b shows the output of the RX comparator for the same frame of RX signal shown in Figure 3a. These signal bursts switches from zero to 5 Volts and reproduces accurately the burst duration of the input signal. Our test modulation scheme based on the length of the burst, despite being ineffective from a practical point of view, allows for checking both the amplitude and time responses of our hardware to the communication signal. First, input level threshold, influence of growing noise levels, and RX dynamic range can be observed from the presence/absence of the comparator output burst for different levels of TX signal, and second, the time duration of the received bursts allows the characterization of the ISI, a quite important parameter to evaluate the maximum communication speed achievable by the system. Regarding this last parameter, in our tests a maximum of one extra pulse at the end of the burst has been observed (as the one shown in the second burst of Figure 3b) at the output of the RX comparator, indicating a low ISI. On the other side, an enormous dynamic range has been observed for the RX amplitude (from 0.8 Volts to 25 Volts). These results suggest the possibility of exploring digital modulation schemes that will lead to propagation speeds over 200 kbps with a 1 MHz carrier (depending on the spectral efficiency of the modulation), or even higher if the carrier frequency is increased. Figure 3c shows the recovered digital frame with alternating 1,s and 0,s after the firmware demodulation of the output comparator signal Figure 3b. Note that symbol duration is the same for “1” and “0” (13 us), but at the stage where the signal in Figure 3c is taken the demodulated digital value is available before for “1” then for “0”, and this is the reason why the pulses with level “1” are longer than the ones with level “0”. When the digital frame is extracted out the bit duration becomes equal for 1 and 0.

As a summary, Table 1 shows the prototype communication parameters, and a picture of the real circuit can be seen in Figure 4. The design here described, has been integrated into the same printed circuit board (PCB) in the I-V tracer designed in our previous work [4], and for this reason not all the components are used or installed in the PCB.

## 4. Discussion

Within this paper, as a natural continuation of our previous paper [16] where a simple loop communication theory was exposed and tuning circuit was designed, a low-cost practical implementation of this ring topology PLC has been fully developed. An auspicious performance is observed from this PLC system governed by loop resonance. The modulated signal is successfully demodulated and digitally recovered at a 1 MHz carrier, along both communication directions. The circuitry for the implementation of transceivers is extremely simple with a very low-cost microcontroller as the central component and a few external components where the main workload is imposed on the firmware. The configuration described has successfully worked in our TX-RX tests showing a very good performance despite its simplicity and low cost.

One of the novelties presented is the anti-saturation coil designed to avoid the core saturation related to high DC currents in conventional coils or transformers. This solution resolves saturation effects because of the high DC currents present in PV strings. We have used this component in our system as an AC blocking coil, but its application can be extended to many other situations presented in the previous literature (coils or transformers), or even applications out of the PV scope. An autoresonance frequency has been observed for this anti-saturation coil that could find application in transmission/reception transformers where the primary or secondary binding is made from our proposed component.

The cost of each PV module board amounted to 5 Euros, and adding the combiner box circuitry (communications board, loop tuning circuit, inverter blocking coils, and bypass capacitors), the cost attained is about 12 Euros per PV module for a 11 modules string, that was not lower than the lowest cost seen in previous literature (4.52 Euros) per PLC board but surpassed significantly the transmission rate performance of previous studies with favorable signal attenuation. Table 2 shows a comparison of the different PV PLC systems found in the literature relative to ours regarding the cost and communication speed achieved (where indicated) or in terms of carrier frequency. The intent is to integrate this developed PLC system with our I-V curve tracing strategies found in [4]. Further studies will be done to characterize the loop and ascertain the performance of its topology over a range of frequencies and varying electrical lengths.

## 5. Conclusions

After the analysis of previous works about PV PLC communications we found two main coupling methods for injecting the signal in the power line:

Capacitive: where the Transceiver is connected in parallel with the PV module and the signal is coupled to the line through a capacitor.

Inductive: where a transformer is usually connected in series with the PV module and the signal is coupled inductively.

In the first case, some of the proposals ignore the connection of an AC blocking coil in series with the PV module, and this leads to the signal flowing across the PV module, and the AC impedance of the communication path to depend on the working point of the module. In the maximum power point, a module is just between the region of very high dynamic impedance (the upper flat part of the curve) and very low dynamic impedance (the quasi-vertical part on the right of the curve), and any deviation or asymmetry in one module can push the working point to the second region, and the corresponding transceiver will be short-circuited, preventing the communication to work. If a blocking coil is mandatory to solve this problem (in parallel capacitive coupling) and a transformer is connected in series for inductive coupling, we will have the entire DC string current flowing through a coil in both cases, and with it the aforementioned saturation problem.

The first novelty presented in this paper is the anti-saturation coil, that affords our circuitry the ability to overcome the drawback above described in a very simple way, without the need of venturing into the use of special or big (and expensive) coil cores. This technique is even applicable to some of the literature configurations that initially ignored the saturation issue.

The other main novelty is to take advantage of the ring topology established, closing the AC signal path to work in a spatial resonance condition as described in our previous work [16]. This topology improves drastically the communication signal levels and considerably reduces the differences between them found in open lines.

Based on these two main improvements, we have designed and successfully tested a very simple and low-cost hardware that improves upon most of the communication speeds previously reported with a cost lower than most of the previous works analyzed (Table 2).

## Figures and Tables

**Figure 1 sensors-22-09207-f001:**
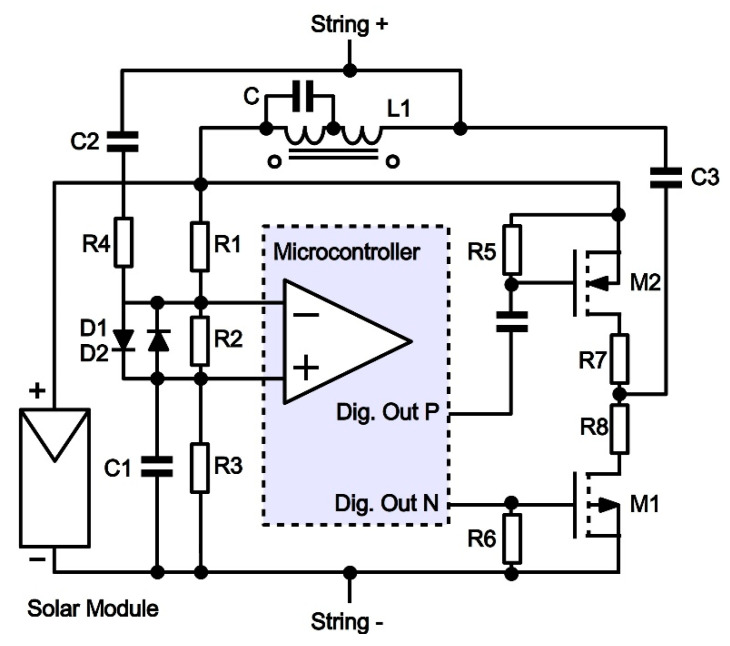
PLC circuitry for transmission and reception.

**Figure 2 sensors-22-09207-f002:**
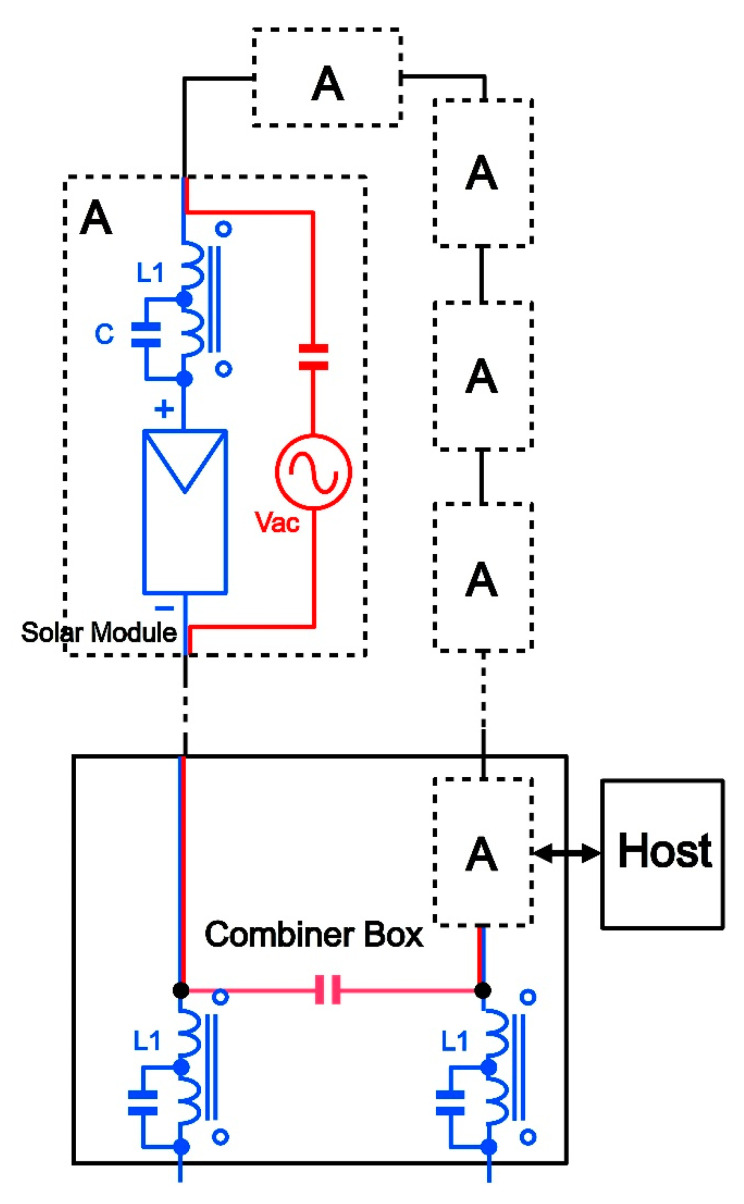
General setup for field measurements.

**Figure 3 sensors-22-09207-f003:**
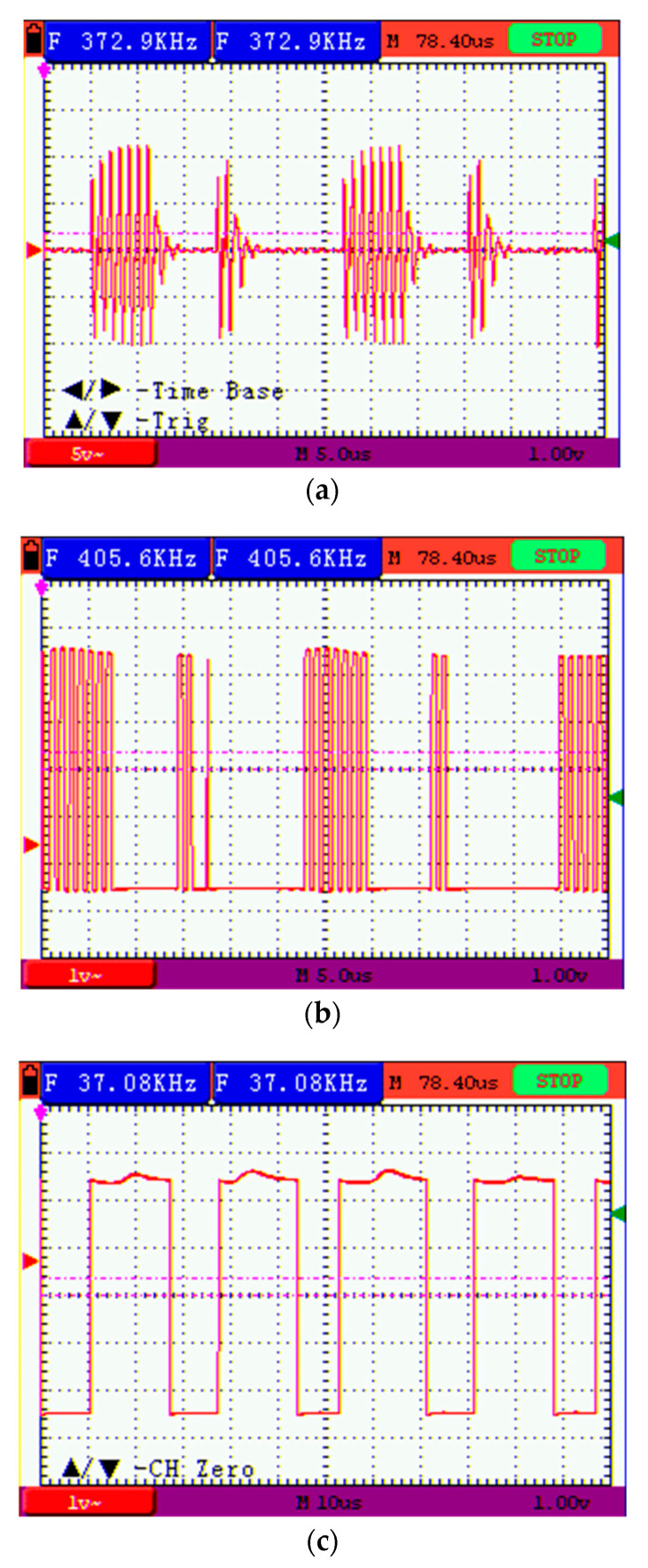
(**a**) Analog form of the modulated signal at one receiver input; (**b**) digital output of the RX comparator for the same received signal; (**c**) digital frame received (alternating 1,s and 0,s).

**Figure 4 sensors-22-09207-f004:**
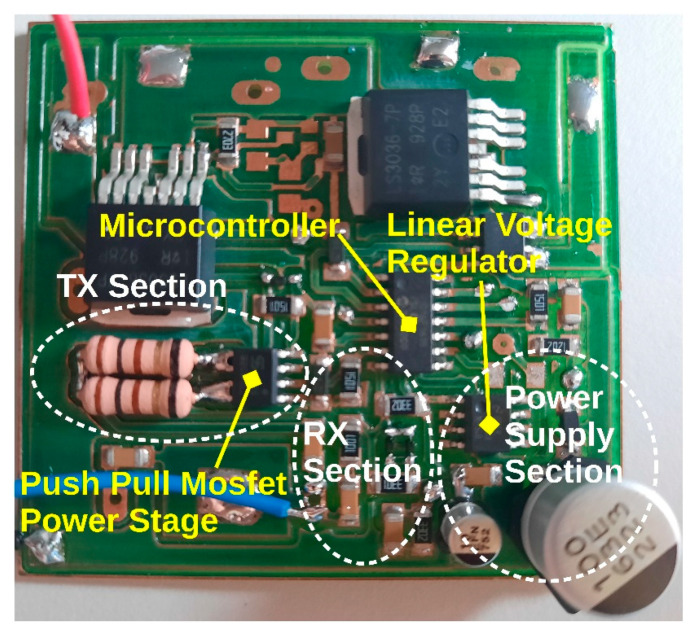
Picture of the real circuit used for testing.

**Table 1 sensors-22-09207-t001:** Prototype communication parameters.

Parameters	Value	Units
Max TX speed ^1^	143	kbps
Bit Error Rate (BER) at max. speed	3 × 10^−3^	-
Noise to Carrier Ratio (NCR) ^2^	31	dB
Inter Symbol Interference (ISI) ^3^	2	us
Energy consumption by bit = 1 transmitted ^1,4^	4 × 10^−10^	Wh
Energy consumption by bit = 0 transmitted ^1,4^	1 × 10^−10^	Wh

^1^ Achieved with the Digital Pulse Wide Modulation Scheme explained in the text, and coding “1” as four carrier cycles long burst, and “0” as one carrier cycle long burst. Symbol duration is seven carrier cycles anyway. ^2^ Measured at RX comparator input in the worst case (Transmission from combiner box towards modules) and the maximum noise measured in our plant (100 mVpp). ^3^ Expressed as time between the end of a burst and the oscillation attenuation of −20 dBV. ^4^ Calculated in the worst case (Transmission from combiner box towards modules).

**Table 2 sensors-22-09207-t002:** Comparison of the different PV PLC systems found in the literature with ours regarding the cost and communication speed achieved.

Author	Cost per PV Module	Carrier Frequency	Bitrate
Napoli et al. [6]	30 Eur	132.5 KHz	n/a
Ochiai et al. [7]	>25 Eur (*)	1 KHz	n/a
Sanchez et al. [8]	10 Eur	100 KHz	n/a
Han et al. [5,9]	>25 Eur (*)	2–30 MHz	4 Mbps
Mao et al. [12]	5 Eur (**)	200 KHz	10 Kbps
Daldal et al. [14]	>70 Eur (*)	n/a	n/a
Proposed method	12 Eur	1 MHz	200 Kbps

(*) Not specified. Estimated from the components described. (**) Coupling coil needed in series with the string, possibility of saturation.

## Data Availability

Not applicable.

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
