# Peer review of "A Power-Line Communication System Governed by Loop Resonance for Photovoltaic Plant Monitoring"

_sensors, 2022, doi:10.3390/s22239207_

Round 1
Reviewer 1 Report
In this paper low-cost and extremely simple transceivers intended to be installed within each PV module of a string have been designed and successfully tested. In addition, an anti-saturation coil has been conceived to avoid saturation of the core when the entire DC of the string flows through it. It’s an interesting work, however, I have the following concerns:
· Most references are very old, please add the latest papers in the introduction section.
· References numbering is anonymous as [2] and [3] are given after the [2]-[4], please arrange the reference numbering according to the standard format.
· Some sentences are senseless such as in line 91, “ [8] employed pulse modulation as well but instead in the voltage domain through capacitive coupling for a low AC impedance transmission path” So, check the whole paper properly to update considering the grammar mistakes.
· In line 265, why to use the anti-saturation coil with the PV module, please give some details.
· In the results, the cost and performance comparison with the existing work is not presented. Please provide the comparison for the validation.
· In line 331, how it is validated that the proposed model has less cost as compared to the available models.
· The conclusion section should be added to the paper to conclude the findings.
Author Response
Thank you for your suggestions, the answers are in the attached file.

Reviewer 2 Report
Congratulations for the work done. I have some questions regarding what you have written.
In line 172, you wrote "C is the capacitance...". Please indicate in figure 1 or in the text which capacitance is C.
There is no uncertainty discussion of results.
Discussion must be rewritten in form of conclusions.
Were the measurements carried out while the plant was in operation, or in the absence of power supply?
Author Response

(The authors gave the same response as above.)

Round 2
Reviewer 1 Report
All comments are addressed and look better now.
Reviewer 2 Report
Corrections are well done